# Pull-Up Performance Is Affected Differently by the Muscle Contraction Regimens Practiced during Training among Climbers

**DOI:** 10.3390/bioengineering11010085

**Published:** 2024-01-17

**Authors:** Laurent Vigouroux, Marine Devise

**Affiliations:** ISM (Institute of Movement Sciences), CNRS, Aix-Marseille University, 13288 Marseille, France; marine.devise@univ-amu.fr

**Keywords:** pull-up capabilities, power, climbing, training

## Abstract

Sport climbing performance is highly related to upper limb strength and endurance. Although finger-specific methods are widely analyzed in the literature, no study has yet quantified the effects of arm-specific training. This study aims to compare the effects of three types of training involving different muscle contraction regimens on climbers’ pull-up capabilities. Thirty advanced to high-elite climbers were randomly divided into four groups: eccentric (ECC; *n* = 8), isometric (ISO; *n* = 7), plyometric (PLYO; *n* = 6), and no specific training (CTRL; *n* = 9), and they participated in a 5-week training, twice a week, focusing on pull-ups on hangboard. Pre- and post-training assessments were conducted using a force-sensing hangboard, analyzing force, velocity, power, and muscle work during three pull-up exercises: pull-ups at body weight under different conditions, incremental weighted pull-ups, and an exhaustion test. The CTRL group showed no change. Maximum strength improved in all three training groups (from +2.2 ± 3.6% to +5.0 ± 2.4%; *p* < 0.001); velocity variables enhanced in the ECC and PLYO groups (from +5.7 ± 7.4 to +28.7 ± 42%; *p* < 0.05), resulting in greater power; amplitude increased in the ECC group; and muscle work increased in the PLYO group (+21.9 ± 16.6%; *p* = 0.015). A 5-week training period effectively enhanced arm performance, but outcomes were influenced by the chosen muscle contraction regimens and initial individual characteristics.

## 1. Introduction

Over the past two decades, rock climbing has witnessed a substantial rise in popularity, reaching an estimated 45 million climbers worldwide in 2019, as indicated by the International Federation of Sport Climbing (IFSC). It reached its pinnacle with its introduction at the Tokyo Olympics in 2021, featuring a combination of the three sporting disciplines: bouldering, lead, and speed. In the upcoming 2024 Paris Olympics, climbing will be divided into two contests, speed climbing on one hand and a combined bouldering and lead climbing contest on the other. Sport climbing performance requires the combination of complex specific skills, such as fine techniques, psychological resilience, and physiological capabilities [1]. Prior research extensively delved into the former, with a majority of studies concluding that expert characteristics are closely related to strength and endurance in the upper limbs [2]. Notably, climbing involves sustained intermittent isometric contractions, demanding fingertip strength for contact with small holds and resistance to fatigue in the forearm and hand muscles [3]. Performance also demands substantial explosive strength and power developed by the arms, facilitating upward movement from one position to the next, and endurance to fatigue induced by successive arm lock-offs and dynamic climbing movements (e.g., dyno). Consequently, a significant portion of training efforts by both trainers and climbers is directed towards enhancing the ability of fingers, forearms, and arms. While finger-specific assessment and training methods have undergone comprehensive analysis in the existing literature [4,5,6], limited research has focused on assessing and quantifying the training effects of the physiological capabilities of muscles acting at the elbow and shoulder (collectively referred to in this study as arm muscles).

To dynamically assess and train climbers’ arm muscles, pull-up movements have emerged as the most prevalent exercise and have been the subject of scientific research for a couple of decades. They allow the measurement of several key variables for climbing, including explosive strength, power, and endurance of arm muscles. According to our knowledge, the study by Grant et al. [7] was the first to use successive pull-ups to assess upper limb muscular endurance. Berrostegieta et al. [8] later proposed assessing upper limb power with single traction in climbers, further re-developed as the power slap test [9]. The integration of new technological tools, such as accelerometers, force sensors, and motion capture systems [9,10,11,12,13], has enhanced the reliability of pull-up tests. Many of these studies have demonstrated the relationship between upper limb strength, power, and endurance, and the climbing grade level, thus underscoring the importance of assessing and training these capacities for climbing performance [2,3,10].

Although the pull-up movement may appear relatively simple, it is actually a multi-joint, closed-chain, and upper-body resistance exercise involving a complex interplay of factors. Firstly, one must flex the elbows while extending the shoulders using the dynamic concentric capabilities of the arm and shoulder muscles [14]. This allows the body to move from a hanging extended position to one where the chin is above the hands. These concentric capabilities can be characterized using the force–velocity (F-V) relationship [13] with typical variables including the maximal theoretical force, velocity, and power. This F-V profile is developed through incremental weighted pull-ups and offers the possibility to quantify the one-repetition maximum (1-RM; i.e., the highest amount of weight that an individual can lift during one repetition) in addition to these variables. Secondly, enhanced performances are observed when consecutive pull-ups are performed, which can be explained by both the contribution of the muscle stretch-shortening cycle (SSC) in the arm muscles and the body coordination [15]. More precisely, executing multiple pull-ups involves lower limbs and trunk movements that generate momentum, thus enhancing the overall performance [16]. At the same time, the arm muscles benefit from the prior pull-up’s stretching and activation that restores energy during the upward phase. Devise et al. [10] proposed to assess these former capabilities (body coordination and SSC) by analyzing several types of pull-ups, requesting participants to perform pull-ups with and without lower body coordination, with and without the preceding stretching phase. Interestingly, this study found a correlation between pull-up strength and power and the climber’s grade level only when a pull-up test was conducted with body coordination. This highlights the importance of body coordination skills in climbing. Additionally, fatigue significantly impacts performance, influencing both the number of successful repetitions and the power developed during the end of a series of pull-ups. The capacity to withstand fatigue can be assessed by measuring the muscle work and the maximum number of pull-ups performed during an exhaustion test [10,17].

Pull-ups can thus be considered as (semi)sport-specific tests, with the same performance factors required for sport climbing. Alongside using pull-ups as a testing exercise, trainers also use them in training sessions. Various pull-up exercise modalities can be implemented on either a campus board or a fingerboard, depending on the muscular contraction regimen, including concentric, isometric, and eccentric contraction regimens. In practice, climbers employ different pull-up techniques, such as locking their arms at given angles (isometric training), performing rebounds (plyometric training, i.e., quick movements involving an eccentric contraction followed immediately by an explosive concentric contraction), and slowly executing the downward phase of the movement (eccentric training). Modulating the exercise intensity by adding or subtracting weight to the climbers is also achievable. Nevertheless, little is known about the real benefits of these methods in relation to the capabilities previously defined. The studies on arm training conducted so far have incorporated exercises that engage fingers as well [18,19] thereby making it difficult to distinguish between the impact on the fingers and the impact on the arms. As a result, proposals and guidelines for arm muscle development remain empirical or have been adopted from other sports activities.

In other sports movements such as drop jumps with lower limbs, the different regimens of strength training (either concentric, plyometric, eccentric, or isometric) have been widely investigated in the literature and the results have shown different magnitudes of adaptation to muscle hypertrophy, strength, and power [20,21,22]. In particular, eccentric training demonstrates greater effects on strength, hypertrophy, and power compared to traditional resistance training (i.e., concentric training) [22]. The same can be said for isometric training, but it involves highly angle-specific adaptations and limited transfer to other muscle lengths [23,24], whereas plyometric training is favorable for developing the ability to produce maximal force with velocity [20]. Nonetheless, the benefits of each regimen are controversial and highly depend on the specific movement and/or assessment used during testing. This is mainly due to the principle of specificity, which states that the body adjusts to the particular demands placed upon it. Consequently, only generic knowledge can be inferred from prior research on lower limbs and applied to pull-up exercises. There is, therefore, a need to assess each training modality’s advantages for pull-up performance.

The aim of this study was to compare the effects of three different types of training involving different muscle contraction regimens (isometric, eccentric, and plyometric), commonly practiced by climbers, on the pull-up capabilities (i.e., the concentric muscular capacities, the ability of body coordination, the capabilities of the SSC, and the capacity to resist fatigue). Following a 5-week training period, four groups of participants (isometric-, eccentric-, plyometric-based training groups, and a control group) were compared. The benefits of training were measured through a battery of tests aimed at characterizing the F-V relationship, the contribution of body coordination, the capabilities of a stretch-shortening cycle, and arm endurance. We hypothesized that training would enhance power developed during pull-ups, through different mechanisms depending on the muscular contraction regimen employed. More specifically, eccentric training would increase muscle strength, while plyometric and isometric training could enhance force development velocity. We also hypothesized that this training would change the slope of the F-V relationship by modifying the maximum force and velocity and that all types of training would improve endurance.

## 2. Materials and Methods

### 2.1. Participants

A total of 41 male climbers participated in this study. Inclusion criteria were to (a) have a minimum Redpoint climbing level of “advanced level” according to the International Rock Climbing Research Association (IRCRA) scale [25], (b) have more than 2 years of climbing experience with at least 2 sessions per week (indoor, outdoor, climbing-oriented weight training, etc.), (c) be free of injuries in the past 6 months that would make training and/or climbing inadvisable, and (d) not be involved in an arm-specific training program for 6 months. In total, 11 participants dropped out of the study due to illness, personal issues, or other commitments, and 30 participants completed the training program and were included in the analysis. The 30 participants had a mean (±SD) age of 24.7 ± 6.3 years, with a body mass (BM) of 66.0 ± 7.1 kg, a height of 177.7 ± 6.1 cm, and an IRCRA level of 23.1 ± 2.4. All participants practiced both climbing subdisciplines (lead and bouldering) to ensure a representative sample of athletes for the upcoming Olympic contest, which is a combined bouldering and lead climbing contest. Each Redpoint IRCRA level was self-reported for bouldering and lead climbing, based on the best performances over the previous six months. Levels achieved in both disciplines were compared and the highest was retained for the analysis. Prior to the start of the experiment, they were fully informed of the experimental process and signed an informed consent form. The protocol was approved by the national ethics committee (CERSTAPS).

### 2.2. Procedures

An overview of the procedures (i.e., testing and training order) is presented in Figure 1. Participants were randomly divided by a manual method into 4 different training groups: eccentric-based (ECC; *n* = 8), isometric-based (ISO; *n* = 7), plyometric-based (PLYO; *n* = 6), and no specific training (CTRL; *n* = 9). Initially, there were 10 participants in the ECC group, 11 in the ISO group, 11 in the PLYO group, and 9 in the CTRL group, but several participants dropped out during the experiment. The experiment consisted of 5 weeks of training twice a week based on pull-up exercises executed on a hangboard. The week before starting the training (pre-test session) and the week after the final training session (post-test session), climbers were tested to evaluate the benefits of the training intervention. One week prior to the pre-test, participants performed a familiarization session which consisted of executing the same tasks as those in the testing sessions. This enabled them to become accustomed to the tool and the different types of pull-ups described below whilst preventing the occurrence of learning effects. Furthermore, it facilitated the accurate adjustment of the adequate load for the incremental weighted pull-up test. Participants were asked to abstain from training and climbing the day prior to the testing sessions and to be ready to perform to the best of their ability. All climbers were instructed to maintain their usual and regular climbing activities outside of the study. The same experimenter attended the familiarization sessions and all pre- and post-training tests to check the correct task execution and to verbally motivate the participants to ensure maximum performance. The experiment took place away from the competition period.

### 2.3. Materials

The testing exercises and training were performed using SmartBoard (ScienceForClimbing, Peypin d’Aigues, France), a hangboard fitted with force sensors (accuracy 0.8 N, 50 Hz acquisition, 0–4000 N measurement range) measuring the vertical force applied to the holds. The associated app provided visual instructions to guide the participants during the tasks. The largest holds (jugs) were used for all exercises in order to avoid limiting arm capabilities by finger strength and endurance [26]. Force data were recorded during each exercise and then exported for post-acquisition analysis.

### 2.4. Pre- and Post-Test Sessions

At the beginning of every test session, each participant underwent a 20 min standardized warm-up routine consisting of upper limb muscular awakening exercises (dynamic stretching with an elastic band, shoulder and wrist rotations, scapular retractions, etc.) followed by a few easy climbs and traverses. Participants ended the warm-up by performing one pull-up in each of the experimental testing exercises detailed below (4 pull-ups in total, with a 2 min rest between each). The testing sessions followed the previously proposed methodology by Devise et al. [10]. Briefly, they were composed of three blocks of exercises conducted to assess the overall pull-up capabilities of the participants: the jump tests, the incremental weighted pull-ups, and the maximum repetition exercise. In the first block, the exercises involved performing a single, explosive two-armed pull-up with maximum velocity and power. Three modalities were tested to determine different aspects of muscular and body coordination using pull-ups: the Strict pull-up only employed arm use to pull the body up, determining the concentric muscular capacities; the Normal pull-up allowed for the coordination of legs and hips, measuring the combined muscular and body coordination abilities; the Countermovement pull-up involved a downward phase prior to a Normal pull-up, including the additional contribution of the SSC capacities. The second exercise block was an incremental weighted pull-up test and allowed the F-V relationship to be established. Participants completed between 4 and 6 Strict pull-ups with incremental added loads, until their 1-RM was reached. The third and final block measured muscular endurance, requiring participants to perform an exhaustion test consisting of executing as many consecutive explosive pull-ups as they could until exhaustion. To prevent the effect of fatigue, at least 2 min of rest separated each pull-up, and a minimum of 20 min of rest was respected before the exhaustion test.

### 2.5. Training Sessions

All three training programs used the SmartBoard jugs. These training programs were designed based on training practices commonly used in the climbing community. Each included concentric contractions and a particular muscular contraction regimen (eccentric-concentric or isometric-concentric for example).

The repetitions and sets have been carefully adjusted to optimally correspond to the workload between protocols in regard to effort duration, intensity, and perceived load. This was achieved through a pilot study conducted with 10 participants (who were not included in the present study) and resulted in the readjustment of the protocols based on these preliminary findings.

#### 2.5.1. Eccentric Protocol (ECC)

The ECC group followed an eccentric-based training (Appendix A) which involved executing 3 repetitions of only the downward phase of a pull-up. The participants were instructed to begin at the top position with fully flexed elbows and the chin above the hands. A chair was used to attain the starting position. Then, they controlled their descent with arm action until reaching a fully arm-extended hanging position. They were asked to regulate velocity to achieve a 5 s duration for every downward phase. The participants were loaded at 95% of 1-RM, and each downward phase was separated by 10 s intervals. To optimize the eccentric training effect [27], participants performed 3 repetitions of the downward phase followed by 3 concentric upward phases (jumps) executed at body weight, alternating between Normal and Strict jumps and limiting the eccentric phase by removing the hands from the hangboard at the top of the pull-up and taking off directly on their feet if possible. Six sets of this exercise were asked by session, with 3 min of rest between each set.

#### 2.5.2. Plyometric Protocol (PLYO)

The PLYO group followed a training based on plyometric movements (Appendix A). The participants were asked to execute consecutive explosive plyometric pull-ups at body weight. The plyometric pull-ups consisted of performing small rebounds by briefly removing the hands from the hangboard at the top of each pull-up and retaking the hangboard before the downward phase. The number of plyometric pull-ups was adjusted to each participant, ranging from 7 to 11, depending on their maximum number of pull-ups performed during the exhaustion test, which remained the same throughout the program. This number was self-selected by the participants with the experimenter during their initial training session, with instructions to stop the set when they felt they were losing too much velocity to execute the pull-ups explosively enough. Further repetitions would cause excessive fatigue and would not elicit additional gains [28]. Six sets of this exercise were performed by session, with 3 min of rest between each set.

#### 2.5.3. Isometric Protocol (ISO)

The ISO group followed a training that was centered on isometric lock-offs executed at body weight (Appendix A). In order to maximize the effect of isometric training [24], the lock-offs were performed at different joint angles and were complemented by concentric efforts (jumps). The exercise began with an initial jump, followed by reaching a targeted elbow flexion angle (either 60°, 90° or 120°; 180° meaning full extension) which was held for 7 s, before concluding the pull-up with maximum power. A second jump was then required before reaching the second targeted angle. Finally, participants executed a third jump and reached the third angle, followed by a final jump to complete the series. The type of jumps alternated between Normal and Strict. It was instructed to the participant to limit the eccentric phase by removing their hands from the hangboard at the top of the pull-up and taking off directly on their feet. Six sets were executed by session, with 3 min of rest between each set. The order of the targeted angles was randomized across the sets and the sessions.

### 2.6. Data Analysis

The recorded force data ($\vec{F}$) from the jump tests and the incremental weighted pull-ups (pre- and post-tests) were low-pass filtered (fourth-order Butterworth, cut-off frequency: 3 Hz). Based on Newton’s second law ($\sum \vec{F}=BM.\vec{a}$), acceleration ($\vec{a}$) was determined, and muscle power (P(t) in W·kg^−1^) was calculated as the product of the force (in N·kg^−1^) and the velocity at each instant with P(t)=F(t)×v(t), the velocity (v(t) in m·s^−1^) being computed using a time-integration of the acceleration.

For the jump tests, the execution time for the ascent phase was determined. Then, the data were re-sampled (100 points) to enable comparison and time was expressed as a percentage of the ascent phase of the pull-up cycle. The peak force and the peak power were identified in each condition and their timing as a percentage of the ascent phase duration was recorded. The mean force, power, and velocity of each trial were computed as averages of force, power, and velocity, respectively, at each instant of the ascent phase of the pull-up.

For the incremental weighted pull-ups, the mean force of the ascent phase was expressed as a function of the mean velocity for each trial in order to compute the F-V relationship. A linear regression was used to define the linearity of the relationship (r^2^) and the slope of the F-V relationship for each subject. The regression curves were extrapolated to obtain the theoretical maximum force (F0) and maximum velocity (V0) and correspond to the y- and x-intercepts, respectively, of the curve with the force and velocity axes.

For the exhaustion test, the deducted velocity was high-pass filtered (fourth-order Butterworth, cut-off frequency: 0.3 Hz) to avoid phenomena of noise amplification and error due to successive integrations over time. The number of executed pull-ups was counted and the mechanical work (i.e., energy expended in kJ) was computed.

### 2.7. Statistics

Descriptive statistics (Mean ± SD) were used to present the results of each variable. The statistical tests were processed with the use of the software R. A one-way analysis of variance (ANOVA) was conducted to ensure that there were no significant differences in descriptive characteristics between the groups prior to the training. The effects of training on gain differences were assessed among the four groups for the variables assessed during the Strict, Normal, and Countermovement jump test conditions (i.e., the mean force [Fmean], the maximum peak force [Fpeak], the mean power [Pmean], the maximum peak power [Ppeak], the range of motion of the pull-up ascent phase [ROM]), as well as the variables from the F-V relationship (i.e., the 1-RM, F0, V0, slope) and the variables from the exhaustion test (i.e., the maximum number of pull-ups [Nmax] and the energy expended [Eexp]). A one-way ANOVA was performed when the data assumed homoscedasticity, regardless of the normal distribution (as ANOVA shows little sensitivity to normality), with a Newman–Keuls post hoc test when ANOVA was significant. When the data assumed a normal distribution but not homoscedasticity, a Welch’s ANOVA was performed, with a Games–Howell post hoc test when ANOVA was significant. When the data did not satisfy the normality and homoscedasticity assumptions, a Kruskal–Wallis test was conducted, along with the Dunn post hoc test when the Kruskal–Wallis test was significant. The effect size (η^2^) was computed and defined as small for η^2^ > 0.01, medium for η^2^ > 0.06, and large for η^2^ > 0.14 [29]. The statistical significance level was set at *p* < 0.05.

## 3. Results

The characteristic data of participants in each group are summarized in Table 1 (number of participants according to their preferred subdiscipline, age, body mass, height, redpoint grade, and practice frequency). No significant differences between groups before training were revealed by the ANOVAs.

### 3.1. Jump Test Conditions

All data for each test and participant (without statistical analysis) are included in Appendix A.

Figure 2 summarizes the values in the different parameters obtained during the three jump test conditions (Strict, Normal, and Countermovement tests), as well as the percentage of gain differences before and after the different training programs.

In the Strict jump test, there was a significant main effect of training for Pmean (F(3,26) = 5.80; *p* = 0.004), Vmean (F(3,26) = 4.48; *p* = 0.012), Vpeak (F(3,26) = 3.29; *p* = 0.036), and ROM (F(3,26) = 6.42; *p* = 0.002). The post hoc test revealed that Pmean (+12.0 ± 7.3%, *p* = 0.004), Vmean (+9.7 ± 7.6%, *p* = 0.035), Vpeak (+5.7 ± 7.5, *p* = 0.044), and ROM (+12.1 ± 8.7%, *p* = 0.005) were significantly greater in the ECC group than in the CTRL group. Vmean (+7.7 ± 11.3%, *p* = 0.034) was significantly greater in the PLYO group than in the CTRL group, and a tendency was observed for Vpeak (+6.1 ± 9.8%, *p* = 0.090). For the ISO group, the gain differences ranged from −4.7 ± 7.9 to +0.2 ± 0.8% and were not significantly different from the CTRL group.

In the Normal jump test, there was a significant main effect of the training for the gain differences in Ppeak (F(3,26) = 3.58; *p* = 0.048), with a trend noted for Vmean (χ^2^(3) = 7.39; *p* = 0.060), Vpeak (χ^2^(3) = 7.27; *p* = 0.064), and ROM (F(3,26) = 2.85; *p* = 0.057). The post hoc test revealed that Vmean (+28.7 ± 42.0%, *p* = 0.068) and ROM (+13.4 ± 8.0%, *p* = 0.091) tended to be greater in the ECC group than in the CTRL group. Ppeak (+25.3 ± 23.1%, *p* = 0.078) showed a tendency to be greater in the PLYO group than in the CTRL group. For the ISO group, the changes ranged from −0.7 ± 10.0 to +8.7 ± 15.9% and were not significantly different from the CTRL group.

In the Countermovement jump test, there was only a tendency effect for the gain differences of Fpeak (F(3,26) = 2.54; *p* = 0.079). The post hoc test revealed that for the ECC group, the changes ranged from +1.1 ± 2.3% to +10.4 ± 12.0% and were not significantly different from the CTRL group. For the PLYO group, Fpeak (+10.1 ± 12.3%, *p* = 0.069) tended to be greater than in the CTRL group. For the ISO group, the changes ranged from +0.1 ± 2.3 to +15.6 ± 13.5% and were not significantly different from the CTRL group. No other significant differences were observed in the other variables.

### 3.2. F-V Relationship

The F-V variables (1-RM, Slope, F0, V0) obtained before and after training, as well as the gain differences, are presented in Table 2. The analysis showed a significant effect of training on the gain differences of 1-RM (F(3,26) = 7.66; *p* < 0.001), which ranged from +2.2 ± 3.6% in the ISO group to 5.0 ± 2.4% in the ECC group while it remained stable (−1.5 ± 3.2%) in the CTRL group. The post hoc test revealed that the 1-RM was greater in the ECC, PLYO, and ISO groups than in the CTRL group. No other differences were observed between the training groups and the CTRL group.

### 3.3. Muscular Endurance

The endurance capacity variables (Nmax and Eexp) before and after the training protocols, as well as the gain differences, are presented in Table 3. The analysis showed a significant effect of training for the gain differences of Eexp (F(3,26) = 4.19; *p* = 0.015). The post hoc test revealed that the Eexp was greater in the PLYO group than in the CTRL, ECC, and ISO groups. No other differences were observed between the training groups and the CTRL group.

## 4. Discussion

The aim of this study was to compare the effects of three different types of training, involving different muscle contraction regimens (isometric, eccentric, and plyometric), commonly practiced by climbers, on the pull-up capabilities (i.e., the concentric muscular capacities, the ability of body coordination, the capabilities of the SSC, and the capacity to resist fatigue). As hypothesized, after a 5-week training period, the pull-up capabilities of the climbers who were tested showed improvement in comparison with the control group. More interestingly, the benefits of the training varied among the three types of training.

With regard to the concentric muscle capacities that were evaluated during the Strict jump and the incremental weighted pull-ups, our results showed that all three training groups (the ISO, ECC, and PLYO groups) improved the 1-RM. The higher 1-RM after training indicated that the climbers could lift heavier loads, and their maximum arm muscle strength increased, especially the ECC group, which showed the best improvement (+5.0%). This finding is in accordance with the previous literature on pull-up training that reported similar improvement [28]. This benefit implied a slight modification of the F-V slope by +5.7% (although non-significant) within the ECC group, indicating that the slope became more negative with a slight increase in F0 and no change in V0. Inversely, a slight shift in the F-V slope by −2.2% (likewise non-significant) in the ISO group led to a less negative slope with no change in F0, but a slight increase in V0. Previous literature has shown that explosive-type resistance training improves high-velocity portions of the F-V relationship (i.e., power output at high velocity against a light load) and that heavy resistance training improves the high-force portion of the F-V relationship (i.e., power output at low speed against a heavy load) [30]. These previous conclusions align with the observed F-V evolution trends in our current study since the eccentric modality in the ECC training was executed at 95% of 1-RM and resulted in higher loading of muscles compared with the ISO training, which was executed at body weight in isometric and concentric modes, representing approximatively 55–75% of 1-RM. To summarize the findings on the F-V aspects, our results revealed that a 5-week training program is sufficient to observe some improvements in the concentric muscle capabilities, with greater benefits observed in the ECC group. While the slight changes in the F-V relationship are in line with the literature, it seems that a longer duration or more intense training is necessary to observe more significant modifications. It should be noted that a high interindividual variability in the improvements of the F-V relationship was observed in our study, indicating that each climber may have responded differently to the training, regardless of the specific muscle regimen employed in the training. Hence, individual factors such as initial strength level, climbing level, history, and experience of the climbers could have an influence on the benefit of the tested training [31]. Moreover, it has been shown that other factors such as fatigue, nutrient intake, and sleep can alter the 1-RM daily test with fluctuations around 36% [32], which may contribute to explaining our high interindividual variability. Further research should be conducted to clarify the effects of these individual parameters.

The benefits of training on concentric muscle capabilities were also observable during the Strict jump. Notably, the ECC group increased the mean and peak power at body weight (+12.0% and +8.2%, respectively) due to the higher mean and peak velocity (+9.7% and +5.7%, respectively). The pull-up amplitude was also greater following the ECC training. This point is noteworthy as it suggests that the ECC group that trained across the entire muscle length, from the arms in a close position to their full extension (so with a maximized range of motion), resulted in improved power output on a larger amplitude. This result is in accordance with previous findings observed in lower limbs [33]. In contrast, the gains in concentric muscle capabilities resulting from the PLYO group were more limited. We observed a slightly increased peak power developed by a higher peak velocity (without statistical significance) along with a higher mean velocity. Unlike the ECC group, the pull-up amplitude tended to be reduced in the PLYO group. The probable reason is that during the training sessions in the PLYO group, climbers did not fully extend their arms at the end of the downward phase of the pull-ups. Instead, a slight flexion was maintained to ensure the maximum efficiency of the SSC, but at the expense of limiting the range of motion. Overall, the improvements observed in the ECC and PLYO groups can be attributed to a combination of neural, morphological, and architectural adaptations [27]. Some adaptations may be common across all training types. Particularly, neural improvements result in a greater ability to rapidly recruit larger motor units (i.e., types IIa and IIx), an increase in motor unit firing frequency, and better intramuscular coordination with improvements in synergist coactivation. These neural adaptations have been previously shown in eccentric [27,34] and plyometric [20,21,35] training. Furthermore, the aforementioned studies reported that these training types lead to muscle fiber hypertrophy as well as changes in the mechanical properties and stiffness of the muscle–tendon complex. However, the greater gain in the ECC group could be partially explained by the higher load intensity used compared with the PLYO group. In the PLYO group, an eccentric portion also occurred, but with a lower load (at body weight as opposed to 95% of 1-RM in the ECC group), and it has been found that light eccentric load induced an increase in strength but to a lesser extent compared with higher eccentric load [36]. Similarly, hypertrophy effects are generally lower compared with those induced by heavy resistance training (as eccentric training). At the morphological level, the ECC group could have generated a greater number of sarcomeres in series, which can explain the greater force in a longer muscle length and greater velocity [27]. On the other hand, plyometric training has suggested that this muscle regimen enhances the ability to use the elastic energy and neural benefits of the SSC [21] that produce a positive effect during the concentric phase. This can be observed through increased velocity (peak and mean), which agrees with previous results showing that plyometric training enhances an individual’s ability to rapidly develop force [20,21]. Since our training period lasted only 5 weeks, we can hypothesize that the increase in muscle power first arose mostly from the neural level [27] despite the existence of some evidence that repartition of fiber types (i.e., decrease in the percentage of type IIb fibers [37,38]) and/or architectural adaptations (e.g., increased fascicule length and angle [39]) may also occur within 5 weeks of heavy-resistance training in lower limbs.

The ISO and CTRL groups did not show any significant enhancements in the Strict jump following the training period. The results in the CTRL group indicated that changes in the other training groups are neither attributed to a familiarization effect with the tests nor to other concomitant activities. Our findings in the ISO group are inconsistent with the literature, which reported improvements with this type of training [24]. To increase maximum strength, the review of Lum and Barbosa [24] indeed recommended performing isometric training at 80–100% of 1-RM. However, our ISO training protocol involved climbers blocking during pull-ups at their own body weight (55–75% of their 1-RM). Thus, a suboptimal intensity may explain the lack of significant improvements in the concentric muscle capabilities in the ISO group. Additionally, our ISO protocol was derived from the climbing community’s practice, combining an isometric contraction immediately followed by a concentric contraction. Van Cutsem and Duchateau [40] showed that under these conditions, the maximum rate of torque development can be lower than without preactivation due to a decline in the average discharge rate of single motor units. ISO training may not be efficient in improving performance due to a lack of muscular coordination, which is unable to generate high levels of force.

The Normal jump condition was used to evaluate the concomitant capabilities of concentric muscles with body coordination. One of the main conclusions of our results is that the observed benefits on the peak power are considerably more pronounced in the Normal jump than in the Strict jump. In particular, the PLYO and ECC groups showed a better gain in peak power (+25.3 and +21.1%, respectively, in the Normal jump; +4.7 and +8.2%, respectively, in the Strict jump), whereas the gain remained limited in the ISO group (+8.7% in the Normal jump) and null in the CTRL group. The gain in the peak power in the PLYO and ECC groups can be attributed to higher mean and peak velocity. In the ECC group, the higher pull-up amplitude can also contribute to better performance. For these two types of training, we can, therefore, conclude that in addition to concentric muscle gains, there is a significant improvement in body coordination abilities. The contribution of body coordination during pull-ups is non-negligible, as it was previously shown to enhance power output by +7.3% [10]. This point is crucial as Devise et al. [10] showed that only the variables associated with the Normal jump (thus including body coordination) are correlated with the climbing grade level, whereas a Strict jump (which only mobilizes concentric muscle capabilities) was not. Hence, the ECC and PLYO groups are suitable to contribute to the climbers’ performance from this perspective. Again, the ISO group presented limited benefits for these variables, although it showed more improvement compared with the Strict jump (null in the Strict jump, +8.7% in the Normal jump), suggesting a slight improvement in body coordination, without a gain in the concentric muscle gain.

The SSC in lower limbs is well-described; this phenomenon results in improvements in the performance of a specific movement when it is preceded by a previous downward phase, typically the Countermovement jump [41]. The gains of the SSC arise from the elastic energy that can be stored in muscles, aponeuroses, and tendons during the preceding downward phase and then released during the movement. They are also generated by benefits in the neuromuscular system, with a higher build-up of muscle stimulation and a reduction of muscle slack [41]. In comparison with lower limbs, the SSC has been little studied for the upper limbs. Vigouroux et al. [15] and Devise et al. [10] observed that the use of the SSC during pull-ups can enhance power by +11% and reduce ascent time by −22.3%. In the current study, the Countermovement jump condition was performed to assess the benefits of the three trainings on these phenomena. The main finding revealed that training led to a marginal increase in peak power during the Countermovement jump, amounting to 3.6% for the ECC group and 9.4% for the PLYO group. As similar or greater gains were observed in the Strict jump, it is unclear whether performance during the Countermovement jump is associated with a better SSC performance or with the better concentric muscle capabilities observed during the Strict jump. Therefore, a 5-week training program or our type of training did not appear to be suitable for improving the SSC capabilities. A longer training period is likely to be necessary as previous studies have suggested that significant improvement in the SSC occurred in longer than 5 weeks of training [35]. Nevertheless, it is noteworthy that the ISO group exhibited similar improvement (+15.6% on the peak power) to that of other types of training. Moreover, the ISO group did not show significant enhancement in power from the concentric muscle capabilities during the Strict jump. Thus, it seems that ISO training could be a method to improve the SSC. In the studies on lower limbs, the improved SSC performance with isometric training [42], as well as plyometric and eccentric training [43,44], may be partly due to an increase in the stiffness of the muscle–tendon complex. A stiffer musculotendinous system has also been suggested to optimize SSC performance in the upper limbs during the bench press [45]. Better performance in the Countermovement jump with ISO training could also originate from neural adaptations. In particular, by practicing the ISO training (a combination of isometric lock-off, creating pre-tension, followed by a fast concentric jump), participants may improve their capability to activate their muscles following a preactivation. This complex activation process was highlighted by Van Cutsem and Duchateau [40]. By reducing the muscle slack, stretching the tendinous tissues, and allowing for a quicker force transmission [41], as is the case during the Countermovement jump, muscle activation may be improved. These assumptions should be confirmed through measurements of these phenomena using electromyography and/or neuromuscular investigation.

Finally, with regard to fatiguability, the endurance capabilities of the studied training improved differently. The PLYO group significantly enhanced these capabilities (+32.1%) in comparison with the ECC and ISO groups (+10.1% and +13.0%, respectively). The fact that the PLYO group was required to execute successive explosive pull-ups (between 7 and 11 repetitions) in each series seemed to provide a benefit for resistance to fatigue. Sánchez-Moreno et al. [28] observed similar results in an 8-week training program with a velocity loss of 25% inducing gains in muscular endurance. It also should be noted that they showed that training with a velocity loss of 50% did not improve strength or muscular endurance, emphasizing the importance of high-velocity repetitions over a higher total number. The other training groups repeated a comparable number of pull-ups to the PLYO group, albeit with rest periods in between, which probably led to fewer benefits for enhancing fatiguability. Consequently, we assumed that the impact of fatigue is more influenced by the short rest periods rather than the training muscle regimen. This point is of importance when designing training programs for endurance. The aerobic mechanisms are probably involved in the improvement of local muscular endurance, with a greater skeletal muscle oxidative capacity [46]. Again, additional studies focusing on endurance training are required to expand our understanding of this topic.

In summary, we conducted a study to assess the benefits of different established training types for arm power over a 5-week period. These methods have been frequently employed by trainers and climbers, but until now, their efficacy has remained unquantified. Our study revealed that a 5-week training period based on pull-ups is effective in improving arm muscle power, body coordination, and arm endurance, resulting in significant performance ranging from +5% to +21.9%. The ECC training is particularly suitable to improve power through both concentric muscle capabilities and body coordination capacities. This results in higher pull-up amplitude, corresponding to the demands of climbing which is required to perform in the overall range of motion of the arm joint amplitudes. While the PLYO training also yielded power benefits, it has, however, a tendency to reduce the range of motion, which should be a consideration. On the other hand, the PLYO training is especially suitable for improving endurance, making it applicable for achieving two distinct training objectives (power and endurance improvements). Conversely, the ISO training showed very limited effects compared with the two other training types. A higher loading intensity than the body weight seems required to observe the benefits of this training regimen. Furthermore, given its potential to enhance the SSC, it should be considered to improve neuromuscular optimization.

Limitations must be taken into account when considering our study. First of all, due to several dropouts, the sample size in each group was relatively small. Further studies are required to corroborate our findings. In particular, the benefits are likely to be heavily influenced by each individual and their initial capabilities, suggesting that a larger sample should be tested for each type of training to characterize individual responses and their determinants. Nevertheless, these results represent a first step towards having an overview of the training benefits that are already used by climbers. Direct application of these results can optimize the practice. Additionally, the values obtained with our sample of climbers in the F-V relationship, the jump tests, and the exhaustion test align closely with the prior literature [10,13,28,47,48,49]. This confirms that our sample of climbers is in line with previous studies and provides confidence in the benefits obtained from our training. Further research is also required to investigate how each specific training regimen can be optimized in terms of intensities and repetitions. A second limitation is that while our methodology is suitable to assess the sources of power improvements (concentric muscle capabilities, body coordination, F-V relationship, SSC, and endurance), additional measurements (such as electromyography, echography, etc.) are necessary to establish the underlying factors of the benefits that were observed.

Overall, the training program’s outcomes were dependent on the specific initial characteristics of the individual. This study provided novel quantification and knowledge that is accessible to trainers and climbers to help them optimize their improvements.

## Figures and Tables

**Figure 1 bioengineering-11-00085-f001:**
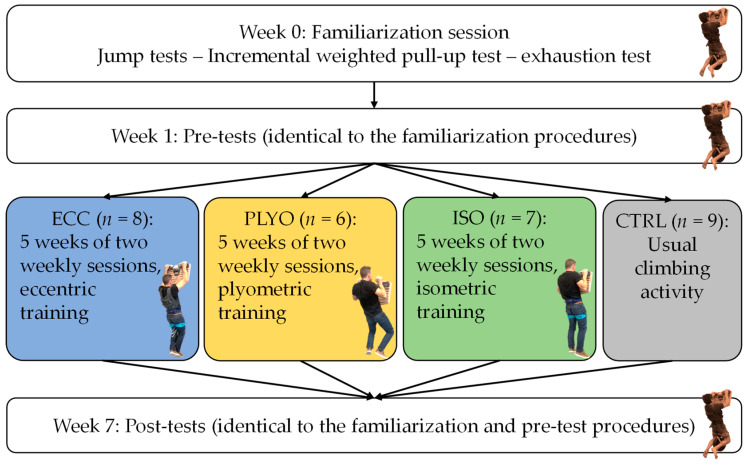
Flowchart showing the study phases and the training intervention.

**Figure 2 bioengineering-11-00085-f002:**
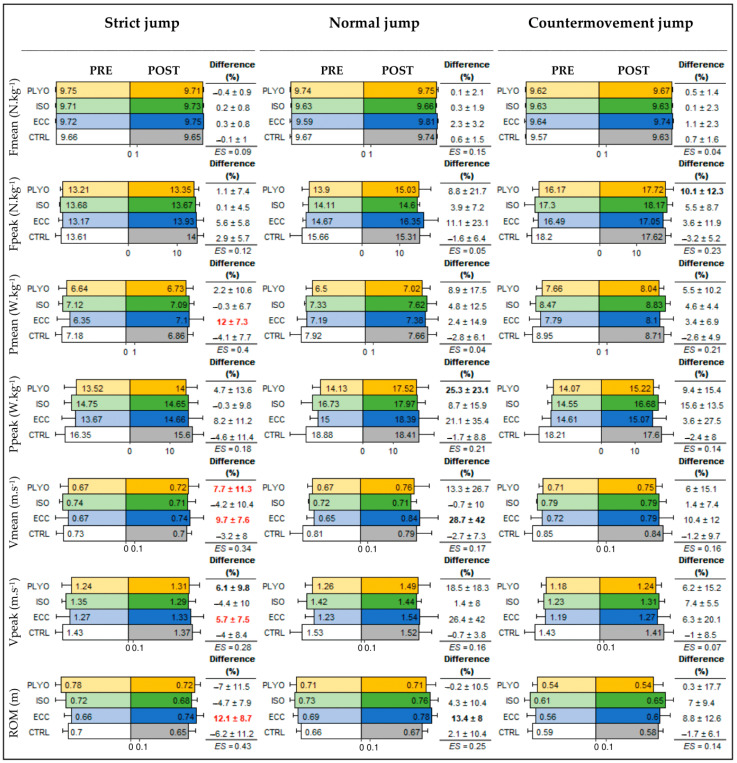
Mean values, mean gain differences (±SD), and effect sizes (ES) of the jump test variables (Fmean—mean force; Fpeak—peak force; Pmean—mean power; Ppeak—peak power; Vmean—mean velocity; Vpeak—peak velocity; ROM—pull-up range of motion) according to the jump form (Strict, Normal, and Countermovement), before (lighter lines) and after (darker lines) each training type (CTRL, ECC, ISO, PLYO). Differences were relative to the CTRL group and are shown on gain differences by red bold text if it was significant (*p* < 0.05) or by black bold text it was a tendency (*p* < 0.10).

**Table 1 bioengineering-11-00085-t001:** Descriptive characteristics (mean ± SD) of the participants of each group (CTRL—control; ECC—eccentric training; ISO—isometric training; PLYO—plyometric training). *p*-values represent the results of the one-way ANOVA comparing the four groups.

	Training Group				*p*-Value
	CTRL	ECC	ISO	PLYO	
Number of participants (discipline preference: boulderers/lead climbers)	9 (4/5)	8 (5/3)	7 (3/4)	6 (3/3)	-
Age (y)	25.7 ± 4.3	26.8 ± 8.4	22.7 ± 6.2	22.7 ± 5.8	0.51
Height (cm)	176.3 ± 5.1	177 ± 8.2	179.5 ± 4.4	178.7 ± 6.4	0.75
Body mass (kg)	67.2 ± 9.0	63.4 ± 6.1	65.7 ± 6.4	67.8 ± 6.7	0.65
Redpoint grade	24.1 ± 2.8	23.4 ± 2.1	22.9 ± 1.5	21.7 ± 2.4	0.28
Practice frequency (sessions/week)	2.7 ± 0.7	2.9 ± 0.7	3.3 ± 1.1	3.0 ± 0.9	0.59

**Table 2 bioengineering-11-00085-t002:** Mean values (±SD), gain differences, and effect sizes (ES) of the force–velocity variables (1-RM—one-repetition maximum; Slope—slope of the force–velocity relationship; F0—theoretical maximum force; V0—theoretical maximum velocity) before and after each training type (CTRL, ECC, ISO, and PLYO).

		Training Group				ES
		CTRL	ECC	ISO	PLYO	
1-RM (kg)	Pre	112.3 ± 17.6	96.4 ± 14.5	102.1 ± 16.8	98.0 ± 9.2	
	Post	111.0 ± 19.6	101.3 ± 15.2	104.3 ± 16.9	101.1 ± 8.3	
	Difference (%)	−1.5 ± 3.2	5.0 ± 2.4 ^a^	2.2 ± 3.6 ^a^	3.2 ± 2.2 ^a^	0.47
Slope	Pre	−12.8 ± 3.7	−10.2 ± 2.2	−11.0 ± 3.8	−9.8 ± 2.4	
	Post	−13.8 ± 5.4	−10.7 ± 2.5	−10.6 ± 3.1	−9.5 ± 1.1	
	Difference (%)	5.9 ± 14.2	5.7 ± 16.1	−2.2 ± 16.0	0.9 ± 21.2	0.04
F0 (N·kg^−1^)	Pre	19.5 ± 3.1	17.2 ± 2.7	18.0 ± 2.8	16.6 ± 2.0	
	Post	19.8 ± 3.9	17.9 ± 2.6	17.9 ± 2.4	16.6 ± 1.1	
	Difference (%)	1.3 ± 4.9	4.4 ± 5.4	−0.4 ± 4.4	1.0 ± 6.4	0.11
V0 (m·s^−1^)	Pre	1.57 ± 0.18	1.71 ± 0.15	1.71 ± 0.3	1.74 ± 0.28	
	Post	1.53 ± 0.28	1.71 ± 0.25	1.76 ± 0.29	1.76 ± 0.17	
	Difference (%)	−3.2 ± 10.0	0.1 ± 10.2	3.5 ± 12.3	3.1 ± 16.7	0.05

^a^ Significantly different from CTRL.

**Table 3 bioengineering-11-00085-t003:** Mean values (±SD), gain differences, and effect sizes (ES) of the endurance variables (Nmax—maximum number of pull-ups; Eexp—total energy expended) before and after each training type (CTRL, ECC, ISO, and PLYO).

		Training Group				ES
		CTRL	ECC	ISO	PLYO	
Nmax	Pre	28.0 ± 8.4	23.1 ± 6.4	21.4 ± 5.9	24.2 ± 4.8	
	Post	29.3 ± 8.4	25.4 ± 7.0	25.7 ± 5.8	27.0 ± 5.5	
	Difference (%)	5.1 ± 7.6	10.1 ± 10.6	21.9 ± 16.6	13.3 ± 21.9	0.18
Eexp (kJ)	Pre	10.7 ± 3.2	8.0 ± 2.4	9.0 ± 2.2	7.4 ± 1.1	
	Post	11.1 ± 3.3	8.8 ± 2.9	10.1 ± 2.6	9.7 ± 1.2	
	Difference (%)	5.1 ± 7.6	10.1 ± 10.7	13.3 ± 21.9	21.9 ± 16.6 ^a,b,c^	0.33

^a^ Significantly different from CTRL; ^b^ significantly different from ECC; ^c^ significantly different from ISO.

## Data Availability

The data presented in this study are available in the Appendix A.

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
