# Peer review of "Pull-Up Performance Is Affected Differently by the Muscle Contraction Regimens Practiced during Training among Climbers"

_bioengineering, 2024, doi:10.3390/bioengineering11010085_

Round 1

Reviewer 1 Report

Comments and Suggestions for Authors

The work seems very interesting.

1.      In the introduction I suggest adding information about the number of climbers in the world or in the country of researchers. This will improve the justification of the topic.

2.      Please justify the selection of only men for the study

3.      L16-127 – ‘’ have more than 2 years of climbing experience with at least 2 sessions per week,’’ - Did the authors set a minimum number of hours per training session or trning sessions per week? Without this information, it is difficult to achieve repeatability of tests

4.      L127 – ‘’ be free of injuries in’’ On what basis were injuries found to be absent?

5.      L129 – ‘’ 11 participants dropped out of the study due to … ‘’ - Describe the initial group allocation and how many people dropped out of each group

6.      ‘’ 2.1. Participants ‘’ - Add sample size calculations.

7.      L133-135 – ‘’ All participants practiced both climbing disciplines (lead and bouldering). Each Redpoint IRCRA level was self-reported for bouldering and lead climbing, based on the best performances over the previous six months.’’

8.      I have the following questions about this part:

a.      Why was it not decided to select a particular sample of boluders or lead climbers?

b.      Do the authors have information on which training (bouldering or lead climbing) the participants spent more time on (in the last six months)? It is difficult to compare groups if, for example, there were people in one group who trained primarily in bouldering and had the best results in bouldering according to IRCRA, compared to another group in which (even by chance) lead climbers trained. This is an additional unexplained variable.

9.      L142 - ‘’ randomly divided’’  - describe how the randomisation was performes.

10.   L236 – ‘’ either 60°, 90° or 120°; 180°’’  - How the maintenance of the 60,90,120 or 180  degree account has been under control?

11.   L287 – ‘’ medium for η²>0.09’’  - There is an error here, it should be 0.06.

12.   Table 1. - Add a comparison of numbers between groups.

13.   Table 1. -  Compare the number of people who declared the best passages in boludering and lead climbing between the groups.

14.   Table 1. - In addition, an important comparison that I miss is that of the number of training sessions. The authors write that ''at least 2 sessions per week'', add information on how much the participants in each group trained on average and compare this between the groups.

Author Response

Thank you very much for taking the time to review this manuscript. Please find the detailed responses below and the corresponding corrections highlighted in red in the re-submitted files.

  1. In the introduction I suggest adding information about the number of climbers in the world or in the country of researchers. This will improve the justification of the topic.

We added this information.

  1. Please justify the selection of only men for the study

There are some differences between men and women concerning the arm muscle power (mermier et al., 2000). To obtain good and robust results in the pull-up movement, we would have needed a representative sample of women in each group, thus about as many women as men, with the same IRCRA level. Unfortunately, it would not have been possible to find as many as we needed in our region. And selecting few women per group would not have been statistically sufficient and would have biased our results towards men.

  1. L16-127 – ‘’ have more than 2 years of climbing experience with at least 2 sessions per week,’’ - Did the authors set a minimum number of hours per training session or trning sessions per week? Without this information, it is difficult to achieve repeatability of tests

We did not set a minimum number of hours per session, because a session could be focused on a climbing-oriented weight training session during 1h or rock climbing for a half-day. The duration can vary while the effectiveness can be similar. As the participants were at an advanced IRCRA level, we can assume that they were able to provide insights into what they considered to be a session, and the inclusion criterium was to practice climbing at least 2 sessions per week. We added information about the type of sessions included in the term of “session”.

  1. L127 – ‘’ be free of injuries in’’ On what basis were injuries found to be absent?

We considered a participant free of injuries if he could train and/or climb without any problem or pain. It was a self-reported information.

  1. L129 – ‘’ 11 participants dropped out of the study due to … ‘’ - Describe the initial group allocation and how many people dropped out of each group

We added the precision.

  1. ‘’ 2.1. Participants ‘’ - Add sample size calculations.

Sample size calculation was done based on a preliminary study. In this study we tested training protocols on 10 participants, divided into 3 groups. This study revealed a pre-post difference of 7% on average power for the 3 groups combined. Our logic was to select the number of participants required to observe significant differences, with an 80% chance of detecting them, as significant at the 5% level, and with a reasonable size effect. Thus, 10 participants were necessary to fulfil this objective. Unfortunately, we started our experiment with 10 participants in each group but many participants dropped out during the training period, leaving us with a smaller sample size. Our samples were determined both theoretically and due to the uncertainty when conducting such a study. We acknowledged this limitation in our manuscript, but we remain confident in our results as they align with our pre-study.

  1. L133-135 – ‘’ All participants practiced both climbing disciplines (lead and bouldering). Each Redpoint IRCRA level was self-reported for bouldering and lead climbing, based on the best performances over the previous six months.’’
  2. I have the following questions about this part:
    1. Why was it not decided to select a particular sample of boluders or lead climbers?

We did not want to select a particular climbing subdiscipline because of the next Olympic contest, which is a combined contest mixing bouldering and lead climbing. The selected athletes will need to practise both disciplines, so we wanted that our participants practised both. Even if the IRCRA level of our sample was not in the very high level, we still think it represented a good sample for the results. We wrote a sentence in this paragraph and in the introduction (about the Olympics).

b. Do the authors have information on which training (bouldering or lead climbing) the participants spent more time on (in the last six months)? It is difficult to compare groups if, for example, there were people in one group who trained primarily in bouldering and had the best results in bouldering according to IRCRA, compared to another group in which (even by chance) lead climbers trained. This is an additional unexplained variable.

Unfortunately, we did not retain the type of training the participants spent more time on in the last six months. However, we followed the type of training or session they did during the study but outside of the experiment sessions. They globally trained more in the subdiscipline in which they had the best results. We can thus hypothesize that the participants spent more time in their best subdiscipline before the experiment.

  1. L142 - ‘’ randomly divided’’ - describe how the randomisation was performes.

The randomisation was made by a manual method. Each participant was assigned to a unique identifier then a random number table was used to allocate participants to different groups. We precised this information in the manuscript.

  1. L236 – ‘’ either 60°, 90° or 120°; 180°’’ - How the maintenance of the 60,90,120 or 180  degree account has been under control?

We informed the participants that 60° corresponded to a locked-arm position, 120° corresponded to an “opened but engaged arm” position and they had to maintain a position in the middle of these two angles, to correspond to 90°. In the first session, we corrected them if necessary, then they were required to maintain these positions for the remainder of the protocol. We provided them with an illustration of each angle to address any potential doubts, and the experimenter check in with the participants midway through the protocol to ensure correct execution.

  1. L287 – ‘’ medium for η²>0.09’’ - There is an error here, it should be 0.06.

We are sorry for the mistake; it has been corrected in the manuscript.

  1. Table 1. - Add a comparison of numbers between groups.
  2. Table 1. - Compare the number of people who declared the best passages in boludering and lead climbing between the groups.

We added a line in the table 1, showing the number of participants in each group (for point 12), with the number of boulderers and lead climbers (for point 13).

  1. Table 1. - In addition, an important comparison that I miss is that of the number of training sessions. The authors write that ''at least 2 sessions per week'', add information on how much the participants in each group trained on average and compare this between the groups.

We completed the table with the ‘practice frequency” line. We did not observed difference between groups.

Reviewer 2 Report

Comments and Suggestions for Authors

The research objective is: The aim of this study was to compare the effects of three common types of training 359 (isometric, eccentric, plyometric) on the pull-up capacities (i.e., the concentric muscular 360 capacities, the ability of body coordination, the capabilities of the SSC and the capacity to 361 resist fatigue).

The research authors must resolve the following points:

1)      The research objectives described in the abstract section, at the end of the introductory section, and at the beginning of the discussion section differ. It is recommended to transcribe exactly the objective described in the discussion section in the rest of the mentioned sections, as it is completer and more comprehensive.

2)      Affirm that the following approach is valid “Sport climbing performance requires the combination of complex specific skills, such 25 as fine techniques, psychological resilience, and physiological capabilities. (Line: 25-26)”, citing some published research, preferably in MDPI journals.

3)      If there is homogeneity in the samples analyzed, according to section 2.7 and table 1, it must be included as one of the inclusion criteria (section 2.1).

Comments on the Quality of English Language

Evaluate with an English language specialist

Author Response

The research objective is: The aim of this study was to compare the effects of three common types of training (isometric, eccentric, plyometric) on the pull-up capacities (i.e., the concentric muscular capacities, the ability of body coordination, the capabilities of the SSC and the capacity to resist fatigue).

Thank you very much for taking the time to review this manuscript. Please find the detailed responses below and the corresponding corrections highlighted in red in the re-submitted files.

The research authors must resolve the following points:

1)      The research objectives described in the abstract section, at the end of the introductory section, and at the beginning of the discussion section differ. It is recommended to transcribe exactly the objective described in the discussion section in the rest of the mentioned sections, as it is completer and more comprehensive.

You are correct, we have revised the sentences and included exactly the same objectives in both the discussion and at the end of the introduction. We could not do the same in the abstract due to word count limitations.

2)      Affirm that the following approach is valid “Sport climbing performance requires the combination of complex specific skills, such 25 as fine techniques, psychological resilience, and physiological capabilities. (Line: 25-26)”, citing some published research, preferably in MDPI journals.

We added a reference, unfortunately no reference on this topic has been published in MDPI journals

3)      If there is homogeneity in the samples analyzed, according to section 2.7 and table 1, it must be included as one of the inclusion criteria (section 2.1).

We are sorry to not fully understand this comment. Our sample of participants were homogenous in terms of participant’s characteristics (mean and SD). Nevertheless, the participants were recruited and were distributed among the group randomly. No other inclusion criteria than those indicated in the manuscript were used.

Round 2

Reviewer 1 Report

Comments and Suggestions for Authors

I accept the authors' answers, I have no further comments. With best regards.